# A Simple, Exact Formulation of Number Counts in the Geodesic-Light-Cone Gauge

Giuseppe Fanizza [1,†], Maurizio Gasperini [2,3,*,†] and Giovanni Marozzi [4,5,†]

1   Instituto de Astrofísica e Ciências do Espaço, Faculdade de Ciências da Universidade de Lisboa, Edifício C8, Campo Grande, P-1740-016 Lisbon, Portugal; gfanizza@fc.ul.pt
2   Dipartimento di Fisica, Università di Bari, Via G. Amendola 173, 70126 Bari, Italy
3   Istituto Nazionale di Fisica Nucleare, Sezione di Bari, 70126 Bari, Italy
4   Dipartimento di Fisica, Università di Pisa, Largo B. Pontecorvo 3, 56127 Pisa, Italy; giovanni.marozzi@unipi.it
5   Istituto Nazionale di Fisica Nucleare, Sezione di Pisa, 56127 Pisa, Italy
*   Correspondence: gasperini@ba.infn.it
†   These authors contributed equally to this work.

**Abstract:** In this article, we compare different formulations of the number count prescription using the convenient formalism of the Geodesic-Light-Cone gauge. We then find a simple, exact, and very general expression of such a prescription which is suitable for generalised applications.

**Keywords:** observational cosmology; covariant averages of astrophysical variables; light cone gauge





Galaxy number counts represent a very useful tool for understanding the Large-Scale Structure (LSS) of our universe and testing the underlying cosmological models. Indeed, the information provided by such a tool on the distribution and overall properties of galaxies can be interpreted as a probe for the associated distribution of dark matter, and thereby as a test of the early cosmological dynamics based on late time observations (see [1–6] and references therein).

One of the first studies involving galaxy number count was presented in a paper discussing the angular auto-correlation of the LSS [7] and its cross-correlation with the anisotropies of the Cosmic Microwave Background (CMB) temperature. Later, this was generalized to include linear relativistic corrections [8–10].

More recently, further developments have been achieved. Among the most relevant, we should mention the computation of the theoretically expected galaxy power spectrum [11] based on the expression of galaxy number counts containing all relativistic corrections (i.e., at the source and observer positions and along the light cone). This advance complements the evaluation of the galaxy two-point correlation function presented in [12]. In addition, ref. [13] computed the "observationally expected" galaxy power spectrum by taking into account effects such as lensing magnification which are important in principle for the evaluation of the power spectrum multipoles, with the results showing that this approach can correct the leading distortion terms due to the redshift at most at the ten percent level.

On the other hand, in a previous paper [14] we presented a new class of covariant prescriptions for averaging astrophysical variables on spatial regions typical of the chosen sources and intersecting the past light-cone of a given observer. In this short communication, we show that with the appropriate choice of ingredients, our integral prescription can exactly reproduce the number count integral introduced long ago as a useful tool in the general context of observational cosmology (see, e.g., [15,16]). In addition, it can reproduce the average integrals recently used in [17,18] as essential ingredients to obtain a reliable (i.e., observationally compatible) prescription for galaxy number counts and for their above-mentioned applications.

Let us start by considering the (possibly realistic) experimental situation in which the sources are not exactly confined on a given space such as a hypersurface (defined for

instance by a scalar field $B(x)$ such that $B(x) = B_s = $ const), and are instead localized inside an (arbitrarily) extended space-time layer corresponding to the interval $B_s \leq B \leq B_s + \Delta B_s$. In thais case, the average is characterized by the following covariant integral prescription [14]:

$$I(\Delta B_s) = \int_{\mathcal{M}_4} d^4x \, \sqrt{-g} \, \rho \, \delta(V_o - V) \Theta(B_s + \Delta B_s - B) \Theta(B - B_s) \frac{\partial^\mu A \partial_\mu V}{|\partial_\alpha A \partial^\alpha A|^{1/2}} . \quad (1)$$

Here, $\rho(x)$ is a scalar field that specifies an appropriate physical weight factor associated with the averaged sources, $V(x)$ is a scalar field (with light-like gradient) which identifies the past light-cone centered on the observer and spanned by the null momentum $k_\mu = \partial_\mu V$ of the light signals emitted by the sources, and $A(x)$ is a scalar field (with a time-like gradient) associated with the following unit vector:

$$n_\mu = \frac{\partial_\mu A}{|\partial_\alpha A \partial^\alpha A|^{1/2}}, \quad (2)$$

which possibly represents a convenient four-velocity reference, but in general depending on the particular observations we are interested in. The particular choice of $\rho$, $A$, and $B$ obviously depends on the physical situation and on the type of observation being performed.

　　Suppose now that we are interested in sources localized between the constant redshift surfaces $z = z_s$ and $z = z_s + \Delta z_s$. In this case, we can choose $B = 1 + z$, where $z$ is the standard redshift parameter defined in general by

$$1 + z = \frac{(u_\mu k^\mu)_s}{(v_\mu k^\mu)_o}, \quad (3)$$

where $u_\mu$ and $v_\mu$ are the velocities of the source and the observer, respectively, not necessarily co-moving in the given geometry, and the subscripts "s" and "o" respectively denote the source and observer positions (see, e.g., [15]). In this case, as we show below, we find that our integral (1) can exactly reproduce the standard number-count prescription of [15–17], provided the fields $A$ and $\rho$ are appropriately chosen, as is shown explicitly just after Equation (14).

　　It is convenient for this purpose to work in the so-called Geodesic Light-Cone (GLC) gauge based on the coordinates $x^\mu = (\tau, w, \theta^a)$ and $a = 1, 2$, where the most general cosmological metric is parameterized by six arbitrary functions $\Upsilon$, $U_a$, $\gamma_{ab} = \gamma_{ba}$, and the line-element takes the following form [19]:

$$ds^2_{GLC} = -2\Upsilon dw d\tau + \Upsilon^2 dw^2 + \gamma_{ab}(d\theta^a - U^a dw)\left(d\theta^b - U^b dw\right). \quad (4)$$

Let us recall here for the reader's convenience that $w$ is a null coordinate ($\partial_\mu w \partial^\mu w = 0$), that in this gauge the light signals travel along geodesics with constant $w$ and $\theta^a$, and that the time coordinate $\tau$ coincides with the time of the synchronous gauge [20]. In fact, we can easily determine $\partial_\mu \tau$ defines a geodesic flow, i.e., that $(\partial^\nu \tau)\nabla_\nu(\partial_\mu \tau) = 0$, which is in agreement with the condition $g^{\tau\tau} = -1$ following from the metric (4).

　　Working in the GLC gauge, we can identify $V = w$; thus, $k_\mu = \partial_\mu w$ and $k^\mu = -\Upsilon^{-1}\delta^\mu_\tau$. Moreover, we can conveniently impose the *general temporal gauge* defined by the condition[1]

$$(v_\mu k^\mu)_o = -1, \quad (5)$$

where the past light cones $w = w_o = $ const are simply labeled by the reception time $\tau_o$ of the corresponding light signals [21], i.e., $w_o = \tau_o$. Hence, in this gauge we have

$$1 + z = \left(u_\tau \Upsilon^{-1}\right)_s \quad (6)$$

and we can replace the $\tau$ integration of Equation (1) with the $z$ integration defined by

$$d\tau = dz \left( \frac{d\tau}{dz} \right) = \frac{dz}{\partial_\tau (u_\tau \Upsilon^{-1})} \, . \tag{7}$$

Note that Equation (7) has to be further integrated, as prescribed by Equation (1), on the spatial hypersurface containing the averaged source. Thus, in order to avoid any confusion between the integrated quantities and the boundaries of the integrals, we omit the explicit subscript "$s$" in the differential integration measure. Finally, for the metric (4) we have $\sqrt{-g} = \Upsilon \sqrt{\gamma}$, where $\gamma = \det \gamma_{ab}$. Thus, our integral prescription (1) takes the form

$$
\begin{aligned}
I(\Delta z_s) &= - \int d\tau \, dw \, d^2\theta \, \sqrt{\gamma} \, \rho \, \delta(w_o - w) \, \Theta(z_s + \Delta z_s - z) \, \Theta(z - z_s) \, n_\tau \\
&= - \int dz \, d^2\theta \, \sqrt{\gamma} \, \rho \, \Theta(z_s + \Delta z_s - z) \, \Theta(z - z_s) \, \frac{n_\tau}{\partial_\tau (u_\tau \Upsilon^{-1})} \\
&= - \int_{z_s}^{z_s + \Delta z_s} dz \, d^2\theta \, \sqrt{\gamma} \, \rho \, \frac{n_\tau}{\partial_\tau (u_\tau \Upsilon^{-1})} \, ,
\end{aligned}
\tag{8}
$$

where we have used the explicit form of $n_\mu$ of Equation (2). It should be stressed here that $u_\tau$ is the time component of the source velocity, while for the moment $n_\tau$ and $\rho$ are both arbitrary variables to be adapted to the physical situation under consideration.[2] Finally, all the integrated functions have to be evaluated on the past light cone $w = w_o$.

We now Consider the number count integral, which evaluates the number of sources $dN$ located inside an infinitesimal layer of thickness $d\lambda$ at a distance $d_A(z)$ and seen by a given observer within a bundle of null geodesics subtending the solid angle $d\Omega$ [15,16]:

$$dN = n \, dV \equiv n \, d\Omega \, d\lambda \, d_A^2 \left( -u_\mu k^\mu \right) . \tag{9}$$

Here, $n$ is the number density of the sources per unit proper volume, $d_A$ is the angular distance, and $u_\mu$ (as before) is the velocity field of the given sources. Finally, $\lambda$ is a scalar affine parameter along the path $x^\mu(\lambda)$ of the light signals such that $k^\mu = dx^\mu/d\lambda$, and is normalized along the observer world-line by the condition

$$\left( v_\mu k^\mu \right)_o = -1 \, , \tag{10}$$

where $v_\mu$ is the observer velocity and the scalar product is evaluated at the observer position [16].

We now move to the GLC coordinates, where $k^\mu = g^{\mu\nu} \partial_\nu w = -\Upsilon^{-1} \delta_\tau^\mu$. The condition $k^\mu = dx^\mu/d\lambda$ then provides the following:

$$\frac{d\tau}{d\lambda} = -\Upsilon^{-1}, \qquad \frac{dw}{d\lambda} = 0, \qquad \frac{d\theta^a}{d\lambda} = 0 \, , \tag{11}$$

and the normalization condition (10) exactly coincides with the general temporal gauge (5). In these coordinates, as anticipated in [22,23], the angular distance $d_A$ satisfies

$$d_A^2 \, d\Omega = \sqrt{\gamma} \, d^2\tilde{\theta} \, ; \tag{12}$$

see Appendix A for an explicit derivation of the above equation. Here, $\tilde{\theta}^a$ represents the angular coordinates of the so-called "observational gauge" [24,25], defined by exploiting the residual gauge freedom of the GLC gauge in such a way that the angular directions exactly coincide with those expressed in a system of Fermi Normal Coordinates (FNC).[3] Using

$$d\lambda = dz \left( \frac{dz}{d\tau} \right)^{-1} \left( \frac{d\tau}{d\lambda} \right)^{-1} \tag{13}$$

and applying Equations (7) and (11), we find that the number of sources of Equation (9) integrated between $z_s$ and $z_s + \Delta z_s$ (as was the case before; see Equation (8)) finally reduces to

$$N = -\int_{z_s}^{z_s+\Delta z_s} dz \, d^2\widetilde{\theta} \sqrt{\gamma} \, n \frac{u_\tau}{\partial_\tau (u_\tau \mathrm{Y}^{-1})} \, . \tag{14}$$

We are now in the position of comparing this result with our previous average integral (8). It is clear that the two are exactly the same, provided the following three conditions are satisfied: (*i*) our scalar density $\rho$ coincides with the number density $n$; (*ii*) the scalar parameter $A$ is chosen in such a way that the unit vector $n_\mu$ coincides with the source velocity $u_\mu$ (and, obviously, $n_\tau = u_\tau$); and (*iii*) the angular directions are expressed in terms of the angles fixed by the observational gauge, i.e., $\theta^a \to \widetilde{\theta}^a$.

It may be appropriate to recall at this point that the number count prescription has been recently used in (apparently) different forms by other authors. For instance, in the context of defining physically appropriate averaging prescriptions the number count has been presented in the following form [17]:

$$dN = n \, dV = \frac{n \, d_A^2}{(1+z) \, H_{||}} \, dz \, d\Omega \, , \tag{15}$$

where $H_{||}$ is a local longitudinal expansion parameter defined by

$$H_{||} \equiv (1+z)^{-2} \, k^\mu k^\nu \nabla_\mu u_\nu \, . \tag{16}$$

Moving to the GLC coordinates and using the temporal gauge (5), we can easily obtain

$$(1+z) \, H_{||} = -u_\tau^{-1} \partial_\tau \left( u_\tau \mathrm{Y}^{-1} \right) . \tag{17}$$

By inserting this result in the definition in (15) and using Equation (11) in Equation (9), we can immediately determine that the number count expressions in (15) and (9) are exactly the same.

As a second example, recall the form of the number-count integral presented in [18] based on the following volume element:

$$dV = \sqrt{-g} \, \epsilon_{\mu\nu\alpha\beta} \, u^\mu dx^\nu dx^\alpha dx^\beta \, , \tag{18}$$

where, as before, $u^\mu$ is the source velocity. Moving to the GLC gauge and projecting this volume element on the light cone $w = \mathrm{const}$, $dw = 0$, we obtain

$$dV = \mathrm{Y} \sqrt{\gamma} \, u^w d\tau d^2\theta \, . \tag{19}$$

Recalling that $u^w = g^{w\nu} u_\nu = -u_\tau \mathrm{Y}^{-1}$ and again using the expression $d\tau/dz$ provided in Equation (7), it is immediately clear that Equation (19) reduces to the same expression of the number count integrand in (14), provided we add the source density $n$ and, as before, the GLC angles are identified with those of the observational gauge [24,25] $d^2\theta \to d^2\widetilde{\theta}$.

The discussion presented in this paper provides a further example of the crucial role played by the GLC coordinates in the simplification and comparison of formal non-perturbative expressions of physical observables. In addition, the simple expression obtained here for the volume element $dV$ in terms of observable variables such as the redshift and observation angles is promising for a number of different physical applications, which will be discussed in forthcoming papers.

We would finally remark that in this brief article we have expressed the galaxy number counts in terms of the redshift. A recent interesting paper [26] has proposed studying galaxy number counts as a function of the luminosity distance of the given sources, rather than their redshift, and showed that there are already differences between the two computational methods at the first perturbative order. Thus, in the near future we plan to evaluate the exact

expression of the galaxy number counts in terms of the luminosity distance by applying the covariant averaging formalism and using the GLC coordinate approach presented in this paper.

**Author Contributions:** Conceptualization, G.F., M.G. and G.M.; methodology, G.F., M.G. and G.M.; formal analysis, G.F., M.G. and G.M.; original draft preparation, G.F., M.G. and G.M.; review and editing, G.F., M.G. and G.M. All authors have read and agreed to the published version of the manuscript.

**Funding:** G. Fanizza acknowledges support by Fundação para a Ciência e a Tecnologia (FCT) under the program *"Stimulus"* with the grant no. CEECIND/04399/2017/CP1387/CT0026, and through the research project with ref. number PTDC/FIS-AST/0054/2021. M. Gasperini and G. Marozzi are supported in part by INFN under the program TAsP (*"Theoretical Astroparticle Physics"*). M. Gasperini is also supported by the research grant number 2017W4HA7S *"NAT-NET: Neutrino and Astroparticle Theory Network"*, under the program PRIN 2017 funded by the Italian Ministero dell'Università e della Ricerca (MUR). G. Fanizza and M. Gasperini wish to thank the kind hospitality and support of the TH Department of CERN, where part of this work has been carried out. Finally, we are very grateful to Gabriele Veneziano for his fundamental contribution and collaboration during the early stages of this work.

**Institutional Review Board Statement:** Not applicable.

**Informed Consent Statement:** Not applicable.

**Data Availability Statement:** Not applicable.

**Conflicts of Interest:** The authors declare no conflict of interest.

### Abbreviations

The following abbreviations are used in this manuscript:

GLC    Geodesic Light-Cone
FNC    Fermi Normal Coordinates

### Appendix A

In order to prove Equation (12), we can combine Equations (3.15) and (3.17) from [23] to express the angular distance in generic GLC coordinates as follows:

$$d_A^2 = 4v_\tau^2 \sqrt{\gamma_s} \left[ \frac{\det\left(\partial_\tau \gamma_{ab}\right)}{\sqrt{\gamma}} \right]_o^{-1} \tag{A1}$$

where, as before, the subscripts "*s*" and "*o*" respectively denote the source and observer positions. We can now move to the observational gauge [24], where we choose a system at rest with the local observer ($v_\tau^2 = 1$) and use Equation (3.16) from [24] to obtain, after a simple calculation of $\gamma_{ab}$,

$$\widetilde{d}_A^2 = \frac{\sqrt{\gamma_s}}{\sin\widetilde{\theta}}, \tag{A2}$$

from which we have

$$\widetilde{d}_A^2 \, d\widetilde{\Omega} = \sqrt{\gamma_s} \, d^2\widetilde{\theta}, \tag{A3}$$

where the tilde denotes the variables of the observational gauges. Finally, note that by using Equations (3.13)–(3.15) from [23] it can easily be shown that the left-hand side of the above equation does not depend on the particular choice of $v_\tau$.

### Notes

[1]   This choice generalises the definition of the temporal gauge, already introduced in [21], such that it can be applied to the case of an arbitrary observer velocity $v_\mu$.

[2]   In our previous paper [14], we applied the above integral in the limit of the small redshift bin $\Delta z_s \to 0$.

[3] The angular directions related to local observations (also used in [15,16]) are indeed those measured by a free-falling observer, and can be identified with the angles of the FNC system [24] where the metric is locally flat around all points of a given world line, with leading curvature corrections (which are quadratic) in the distance.

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
