# Peer review of "A Simple, Exact Formulation of Number Counts in the Geodesic-Light-Cone Gauge"

_universe, doi:10.3390/universe9070327_

Round 1

Reviewer 1 Report

Very nice derivation of galaxy counts in the GLC gauge.

Suggest accept with minor suggestions.

Line 39: why is 'possibly' needed here?

Line 44: why is 'possibly' needed here?

Line 49: Why is 'general' inserted here?

Line 53: 'A and ro are appropriately chosen' Specify what the appropriate choice should be, or refer the reader to later secttions of tthe paper where these choices are beingg made.

References: If at all possible, replace some of the references by the paper's authors with alternative ones.

Author Response

Dear Editor,

we are very grateful to the Referees for their careful reading of the manuscript, their kind comments, and their useful remarks.

We have the pleasure to submit a new version of the manuscript, where we have performed some small changes following the indication of the Referee reports. As requested in its message by the Managing Editor, all changes have been clearly highlighted, in the new version of the paper, by writing the modified text in "boldface" style. 

Here is our answer to the five points raised by the Reviewer 1, and a detailed list of the changes made. 
(Note that the line numbers refer to the original version of the paper).
________________

1) Line 39, after Eq.(1): why is 'possibly' needed here?

The weight factor is indeed to be associated, in some way, to the averaged sources,  and we have thus eliminated the word "possibly". 
_____________

2) Line 44, after Eq. (2): why is 'possibly' needed here?

In that case the word "possibly" is needed, because the ultimate choice of the unit vector n_\mu depends, in general, on the particular observational variables we are interested in. We only need, for the consistency of our formalism, that n_\mu correctly transforms as a vector. We have added a few words to explain this point. 
_____________

3) Line 49, before Eq. (3): why is 'general' inserted here?

Because the source and observer velocities are non necessarily comoving, as usually assumed when defining the redshift. We have added a few words on this point.
_____________

4) Line 53: 'A and rho are appropriately chosen' Specify what the
appropriate choice should be, or refer the reader to later sections
of the paper where these choices are being made.

We have added a comment to refer the reader to the point of the paper discussing the appropriate choices.
_____________

5) References: If at all possible, replace some of the references by the paper's authors with alternative ones.

We are not aware of such alternative references. Hence, in the impossibility of 
removing some of the references made to our own previous work, we have added 6 new references to other scholars' publications, as also suggested in the message received by the Managing Editor, in order to ensure some balance in accordance with standard academic practice. The new six references have been added at the end of the first paragraph of our paper. Let us notice, finally, that it would be probably not appropriate to include a too large, encyclopedic list of references in this paper also in view of the fact that it has to be changed from manuscript type "Article" to manuscript type "Communication", as suggested by the message of the Managing Editor.
_____________

We hope that the improved manuscript may be accepted for publication in Universe.

With many thanks,
Yours sincerely,
G. Fanizza, M. Gasperini and G. Marozzi

Reviewer 2 Report

The paper deals with formulas for galaxy number counts. The formulas are derived using the Geodesic-Light-Cone gauge and compared with several other pre-existing ones. I think the paper is sound and useful and I sugget to publish it on Universe

Author Response

Dear Editor,

we are very grateful to the Referees for their careful reading of the manuscript, their kind comments, and their useful remarks.

We have the pleasure to submit a new version of the manuscript, where we have performed some small changes following the indication of the Referee reports. As requested in its message by the Managing Editor, all changes have been clearly highlighted, in the new version of the paper, by writing the modified text in "boldface" style. 

We hope that the improved manuscript may be accepted for publication in Universe.

With many thanks,
Yours sincerely,
G. Fanizza, M. Gasperini and G. Marozzi

Reviewer 3 Report

The paper describes a comparison of approaches to the cosmic number count distribution and a derivation in the specific geodesic light cone gauge of a general expression for such distributions.  The paper is relatively short and concise.

While this is not my specific area, I did verify a few of the equations the authors derived, sufficient to persuade me that the paper has solid results.  What I can not comment on is the uniqueness of the result -- the geodesic light cone gauge has been described/derived for about a decade, so I would have thought that a number count description would have been published some years ago.   A brief survey of the papers that arise in ADS by searching for 'geodesic light cone gauge' turns up 'number counts' as a topic.  But a 'general' relation does not appear to be among them.  

On that basis, I recommend publication.

Author Response

Dear Editor,

we are very grateful to the Referees for their careful reading of the manuscript, their kind comments, and their useful remarks.

We have the pleasure to submit a new version of the manuscript, where we have performed some small changes following the indication of the Referee reports. As requested in its message by the Managing Editor, all changes have been clearly highlighted, in the new version of the paper, by writing the modified text in "boldface" style.

Concerning the comment of the Reviewer 3, we are aware of the presence in the literature of other papers on number count and GLC gauge but, but we believe that it would be probably not appropriate to include a too large, encyclopedic list of references also in view of the fact that this paper  has to change style, from manuscript type "Article" to the much shorter manuscript type "Communication", as suggested by the message of the Managing Editor.

We hope that the improved manuscript may be accepted for publication in Universe.

With many thanks,
Yours sincerely,
G. Fanizza, M. Gasperini and G. Marozzi